# Gastroprotective Effects of *Ganoderma lucidum* Polysaccharides with Different Molecular Weights on Ethanol-Induced Acute Gastric Injury in Rats

**DOI:** 10.3390/nu14071476

**Published:** 2022-04-01

**Authors:** Baoming Tian, Qin Zhao, Haoyong Xing, Jing Xu, Zhenhao Li, Hua Zhu, Kai Yang, Peilong Sun, Ming Cai

**Affiliations:** 1College of Food Science and Technology, Zhejiang University of Technology, Huzhou 313299, China; tbm2020@zjut.edu.cn (B.T.); 18882022054@163.com (Q.Z.); 18255270010@163.com (H.X.); zhuhuv@sina.com (H.Z.); yangkai@zjut.edu.cn (K.Y.); sun_pl@zjut.edu.cn (P.S.); 2Key Laboratory of Food Macromolecular Resources Processing Technology Research (Zhejiang University of Technology), China National Light Industry, Huzhou 313299, China; 3Longevity Valley Botanical Co., Ltd., Hangzhou 321200, China; sxg012@sxg1909.com (J.X.); zhenhao6@126.com (Z.L.)

**Keywords:** *Ganoderma lucidum* polysaccharides, cascade membrane technology, acute alcohol gastric injury, gastroprotective effects

## Abstract

*Ganoderma lucidum* is known as a medicine food homology that can ameliorate gastrointestinal diseases. To evaluate the gastroprotective effects on different *Ganoderma lucidum* polysaccharides (GLPs), GLP was separated into three parts with different molecular weights using 100 kDa, 10 kDa, and 1 kDa membranes. The mitigation effects of different GLPs on ethanol-induced acute gastric injury were observed in rats. After pretreatment with different GLPs, especially GLP above 10 kDa, the symptoms of gastric mucosal congestion and bleeding were improved; serum myeloperoxidase, inflammatory factor, and histamine were decreased; and antioxidant activity and defense factors (NO and EGF) were increased. Results showed that GLP with different molecular weights had a dose-dependent effect in alleviating alcohol-induced gastric injury. The underlying mechanism might be related to regulating anti-oxidation, promoting the release of related defense factors, reducing inflammatory factors, and reducing the level of histamine in serum. The current work indicated that GLPs above 10 kDa could be applied as natural resources for producing new functional foods to prevent gastric injury induced by ethanol.

## 1. Introduction

The gastrointestinal tract is the largest endocrine organ in the human body, and its mucosal barrier protects it from invasion factors. Gastrointestinal diseases have become a global problem, threatening 5–10% of the world’s population [1]. Ethanol is the most common cause of stomach ulcers and is easily absorbed by the gastrointestinal mucosa, and high concentrations of ethanol damage the stomach lining within 30 min, resulting in gastritis or gastric ulcer symptoms [2]. Severe inflammation will further cause irreversible damage to gastric tissue and gastric cells, which would be a risk of stomach bleeding, stomach ulcer and stomach cancer. Therefore, it is necessary to take some reasonable treatments to protect the stomach [3]. At present, protecting gastric mucosa and reducing gastric acid secretion are effective methods to treat alcohol-induced gastric mucosa injury. Proton pump inhibitors, gastric acid neutralizers, prokinetics, and digestive enzyme preparations are generally used for treatment of gastritis and associated symptoms. However, evidence has suggested that long-term use of these anti-gastritis drugs may be associated with several adverse effects, such as bloating and constipation. In severe cases, chronic atrophic gastritis and vitamin B deficiency, cardiovascular events, headaches, depression, and constipation would be induced [4]. Therefore, the search for a natural active substance is the key to gastric ulcer disease research. Based on previous research, the pro-inflammatory mediators, e.g., reactive oxygen species (ROS), cytokines, and neutrophil infiltration, are the main factors in gastric mucosal injury, while cytokines, e.g., tumor necrosis factor-α (TNF-α), interleukin(IL)-6, and IL-10 play a key role in the occurrence and development of gastric ulcer; the mechanism of alcohol-induced gastric injury has not been fully elucidated [5].

In recent years, many studies have focused on the development of safe and effective functional foods to regulate inflammatory response, especially some natural ingredients. These ingredients have the advantages of high safety and good anti-inflammatory effects; accordingly, they received widespread attention [6,7]. In addition, more studies confirmed the protective effects of natural polysaccharides on gastric ulcers and gastritis. Polysaccharides of *Dendrobium officinale* [Kimura & Migo] could protect the gastric mucosa by inhibiting cell apoptosis induced by oxidative stress [8]. Polysaccharides of *Momordica charantia* could reduce ethanol-induced gastric injury in rats by inhibiting gastritis and oxidative stress [9]. Accordingly, polysaccharides could be used as an effective treatment or health supplement for gastric mucosal diseases. *Ganoderma lucidum*, an important medicinal and food fungus, is mainly distributed in China, Korea, and Japan. Physiologically active substances in *Ganoderma lucidum* such as polysaccharides, triterpenoids, alkaloids, steroids, and ergosterol have been isolated and identified [10]. Among them, *Ganoderma lucidum* polysaccharide (GLP) is one of the main resources for its pharmacological activities. Various biological activities of GLP have been extensively studied, such as anti-inflammatory, anti-tumor, anti-oxidant, anti-diabetic, and immunomodulatory activities [11]. The relationships between molecular weights and activities of polysaccharides have been demonstrated, although this study is still limited. GLPs with larger molecular weights had higher activities of scavenging, 1,1-diphenyl-2-picrylhydrazyl (DPPH) and hydroxyl radicals, better reducing power, and better ability to have a cellular-protective effect on yeast cells from ultraviolet radiation damage [12]. GLPs protected against ethanol-induced acute liver injury in vivo and in vitro via toll-like receptor 4/nuclear factor-kappa B (TLR4/NF-κB) signaling pathway reduced the secretion of inflammatory cytokines (e.g., TNF-α and IL-1β) [13]. GLPs with higher molecular weight had better antioxidant and anti-proliferative activities [14]. Moreover, polysaccharides from *Dendrobium nobile* could alleviate ethanol-induced histological damage, antioxidant activities, and the level of epidermal growth factor (EGF) [2]. To our knowledge, there are few reports on the effects of GLPs with different molecular weights on anti-gastritis.

Gradient alcohol precipitation and column chromatography classification are widely used to classify polysaccharides with different molecular weights [15], which might affect their structure and activities [16,17]. In recent years, membrane technology has been widely used in food industries as a green and efficient classified method [18]. Tang et al. [19] used continuous ultrafiltration technology to separate three polysaccharides from water extracts of *Lentinus edodes* and explored their immunomodulation. Ultrafiltration and a series of chromatographic steps were used to separate and purify the polysaccharides. In this study, ultrafiltration membranes of 100 kDa, 10 kDa, and 1 kDa were used to fractionate the crude polysaccharides of *Ganoderma lucidum*. Effects of GLPs with different molecular weights on repairing gastric mucosal damage would be carried out. In summary, the present results provide evidence for understanding different molecular weights and biological activities of GLPs, which can be applied as potential feedstock for functional foods and dietary supplements.

## 2. Materials and Methods

### 2.1. Materials and Chemicals

The fruiting body of *Ganoderma lucidum* was obtained from Longevity Valley Botanical Co. Ltd. (Jinhua, China). Omeprazole was purchased from Shanghai Yuanye Bio-Technology Co., Ltd. (Shanghai, China). Superoxide dismutase (SOD), glutathione peroxidase (GPX), myeloperoxidase (MPO), nitric oxide (NO), epidermal growth factor (EGF), TNF-α, 1L-1β, IL-18, and histamine (HIS) enzyme-linked immunosorbent assay (ELISA) kits were purchased from Nanjing Jiancheng Bioengineering Institute (Nanjing, China). All other chemicals used in this study were of analytical grade.

### 2.2. Animals

Sprague–Dawley rats (150~180 g) were purchased from Hangzhou QDKR Bioscience (Hangzhou, China) and used for ethanol-induced acute gastric ulcer animal experiments. The rats were housed at a room temperature of 25 ± 1 °C with a cycle of 12 h light and 12 h dark (lights on from 6:00 am to 6:00 pm), received food and water ad libitum, and adapted to the experimental environment for 1 week. The sample was administered to experimental animals by oral gavage for 1 week, and the administration volume was the mass of rats × 0.2 mL/20 g. Omeprazole was used as the positive control group, and the administration concentration was 20 mg/kg. Referring to previous studies [2,13,20], the administration doses of GLPs were 100 mg/kg, 200 mg/kg, and 400 mg/kg in each group. One hour after the last sample administration, except for the blank control group, the groups of experimental animals were given 75% ethanol for modeling. 

### 2.3. Preparation of Polysaccharides

The preparation of GLPs is based on our previous studies by cascade membrane technology [20]. 200 g of *Ganoderma lucidum* was added with distilled water to a beaker at a ratio of 15:1 (mL: g), mixed thoroughly with a glass rod, and extracted at 90 °C for 2 h. The extracts were filtered through filter paper (20 μm), and the residue was re-extracted in accordance with the above conditions. These two filtrates were mixed to be the crude extracts of GLPs. The crude extracts were classified by a membrane with 100 kDa at 1 MPa and 700 rpm, and the retentate was concentrated by rotary evaporation. Ethanol was added to be the volume ratio of concentrate to ethanol at 1:4, sealed, and left overnight to obtain the precipitate. GLPs were divided into three grades with different molecular weights using cascade ultrafiltration membrane technology. The precipitate was redissolved with water, rotary evaporated to remove the organic solvent, and vacuum freeze-dried as GLP100. According to the same procedure, the permeate of 100 kDa was filtered with the 10 kDa membrane to obtain the sample as GLP10, and the permeate of 10 kDa was filtered with the 1 kDa membrane to obtain the sample as GLP1. This specific process is shown in Appendix A.

### 2.4. Purification and Characteristics of GLPs

The procedures of purification and evaluation of the characteristics of GLPs were carried out according to our previous study [20]. Different parts of GLPs were purified by DEAE Sepharose fast flow chromatography. The molecular weight of purified GLP was identified by GPC with an HPLC system (LC-10A, Shimadzu, Japan) equipped with a gel column (BRT105-104-102, China) and a differential detector RI-502. The methylation of GLP was analyzed by gas chromatography–mass spectrometry (Trace 1300/ISQ, Thermo Fisher, MA, USA). Monosaccharide composition was identified by ion spectrometer (ICS1000, Thermo Fisher, MA, USA). The functional groups of purified GLP were identified by Fourier transform infrared spectroscopy (FTIR) (Thermo Scientific™ Nicolet™ 6700, Thermo Fisher, MA, USA) from 4000 cm^−1^ to 400 cm^−1^.

### 2.5. Rat Groups and Experimental Procedure

#### 2.5.1. Ethanol-Induced Acute Gastric Injury Model and Treatment

Rats were divided into groups of blank control, model control, positive control (Omeprazole, 20 mg/kg), GLP100-L (>100 kDa, 100 mg/kg), GLP100-M (>100 kDa, 200 mg/kg), GLP100-H (>100 kDa, 400 mg/kg), GLP10-L (>10 kDa, 100 mg/kg), GLP10-M (>10 kDa, 200 mg/kg), GLP10-H (>10 kDa, 400 mg/kg), GLP1-L (>1 kDa,100 mg/kg), GLP1-M (>1 kDa, 200 mg/kg), GLP1-H (>1 kDa, 400 mg/kg). There were 10 rats in each group. After the last administration, animals were strictly prohibited from eating (no drinking water) for 24 h. Each group of experimental animals was given samples or the same volume of normal saline. One hour later, the model group and sample-treated groups were given 75% ethanol at a dose of 10 mg/kg. The blank control group was given an equal volume of normal saline. One hour later, the animals were euthanized, the entire stomach was exposed, and the pylorus was ligated. The appropriate amount of 10% formaldehyde solution was perfused and fixed for 20 min. An incision was made along the greater curvature of the stomach, and the contents were removed. Lavage of the stomach was performed to unfold the gastric mucosa, and the length and width of the bleeding point or bleeding band were measured with a vernier caliper under a stereoscopic dissecting microscope or the naked eye. A pathological section of the most severely damaged part was taken and scored. 

#### 2.5.2. Rat Gastric Injury Evaluations

The stomach was taken from the euthanized rats immediately and washed with cold physiological saline. The stomach was laid flat on filter paper to observe the size, shape, and occurrence location of the gross gastric ulcer and erosion hemorrhagic points. The ulcer index and ulcer inhibition rate were calculated according to the Guth standard [21]. Observation and scoring: (1) Punctured bleeding: hemorrhagic erosion of small spots or gastric mucosal defects less than 1mm, called punctured ulcer, every 3 punctured ulcers count 1 point; (2) Strip bleeding: use vernier calipers to measure the maximum length diameter of the ulcer surface and the maximum width diameter perpendicular to the maximum length diameter, and calculate the product of the two. If the width is 1 mm, the length per millimeter is 1 score; the width of 2 mm is 2 scores per millimeter; the width of 3 mm is 3 scores per millimeter; (3) The sum of calculated values of punctured and striated bleeding is the ulcer index. Ulcer inhibition rate was calculated according to the formula below.
$$Ulcer\;inhibitory\;rate\%=\left(1-\frac{ulcer\;index\;of\;sample\;treated\;mice}{ulcer\;index\;of\;model\;group\;mice}\right)\times 100\;$$

#### 2.5.3. Histological Analysis

The hematoxylin and eosin (H&E) staining method was used for histopathological observation of the general microstructure of gastric mucosa [22]. The gastric tissue was cut into 5 mm with a scalpel and fixed overnight with 10% neutral buffered formaldehyde solution. Afterword, it was dehydrated and paraffin embedding was performed. It was H&E stained, sealed with neutral gum, and observed under a microscope.

### 2.6. Determination of Biochemical Indexes in Rats

The whole blood of the rats was dissected, placed in an enzyme-inactivated centrifuge tube, and centrifuged for 15 min at 6000× *g* rpm at 4 °C. The upper clear serum was transferred to a new 1.5 mL Ep tube. The sample was stored in a refrigerator at −80 °C for subsequent biochemical indicators and ELISA detection, which was measured according to the manufacturer’s instructions. The antioxidant capacity of rats was evaluated by measuring the content of antioxidant-related enzymes (SOD, GPX, MPO) in serum. The self-healing ability of the rats was evaluated by detecting the content of EGF and NO in serum. The degree of gastric injury was evaluated by detecting the contents of HIS. The degree of inflammation in the rats was evaluated by detecting the content of inflammatory factors (TNF-α, IL-1β, and IL-18).

### 2.7. Statistic Analysis

All data were expressed as the mean ± standard deviation. SPSS16.0 software (SPSS Inc., Chicago, IL, USA) was used for one-way ANOVA. The significant difference between the two groups was determined by the student–Newman–Keuls test. *p* < 0.05 was considered as statistically significant. The significant difference was analyzed by the letter marking method. Origin 2018 software (OriginLab Corp., Northampton, MA, USA) was used for the drawing.

## 3. Results and Discussion

### 3.1. Characteristics of Classified GLPs

Membrane separation as a safe and gentle classification technology was used in the classification of polysaccharides. After ultrafiltration membrane classification, three GLPs with different molecular weights were obtained [23]. In our previous study, the basic characteristics of three purified GLPs were determined [20]. The molecular weights of GLP100, GLP10, and GLP1 were 322 kDa, 18.8 kDa, and 6.4 kDa, respectively, and three GLPs were composed of five monosaccharides: Fuc, Ara, Gal, Glc, Xyl, and Man. Methylation analysis showed that GLP100 has three methylation products, including 2,3,4-Xylp, t-Galp, and 3,4,6-Manp. GLP10 has two methylation products, including 2,3,4-Manp and 2,3,4-Galp. GLP1 has two methylation products, including 2,3,4-Xylp and 2,3,4-Galp. Moreover, three GLPs were β Configuration polysaccharide.

### 3.2. Appearance of Gastric Mucosal after GLPs Treatment 

Gastric mucosal injury is characterized by gastric mucosal erosion, extending to the muscle layer to puncture the gastric wall [4]. The ethanol-type gastric mucosa injury model is an acute gastric mucosal injury model. Ethanol can act directly on the gastric mucosa, leading to erosion, bleeding, perforation, and other injuries [22]. It can be judged whether inflammation occurs by observing the gastric mucosa of rats. The stomach wall of normal rats is smooth and elastic, and there are no erosions or bleeding spots. When the gastric mucosa was damaged by ethanol, it caused inflammation, infiltration, prominent oedema, hemorrhage, and obvious diffuse hemorrhage in the submucosa of the gastric mucosa [24]. Polysaccharides, such as that from *Hericium erinaceus* and *Chinese Iron Yam*, have a protective effect on gastric injury [3,24]. By observing the macroscopic map of the gastric mucosa, it was found that after alcohol gavage, pathological symptoms such as mucosal edema and glandular congestion appeared in the stomach. Before alcohol gavage, using *Bletilla striata* polysaccharide for pretreatment, the symptoms of gastric injury were alleviated. However, as with other studies on the anti-gastritis effect of polysaccharides, this study only focused on the improvement effect of different polysaccharides on abnormal lesions of gastric mucosa.

In our study, the gastric mucosa of normal rats had no symptoms of erosion and congestion, as shown in Figure 1A. However, rats modeled by gavage with 75% ethanol showed significant gastric mucosal lesions, congestion, and bleeding (Figure 1B). In Figure 1C, the gastric mucosa of the positive group had only a few hyperemia spots. Compared with the model group, the symptoms of gastric mucosal injury were reduced in the GLP-treated groups. The rats in groups of GLP100-H (Figure 1D) and GLP10-L (Figure 1I) had only slight congestion and a few blood clots in their stomachs. The symptoms of hyperemia and bleeding in groups of GLP100-L (Figure 1F), GLP1-H (Figure 1J), GLP1-M (Figure 1K), and GLP1-L (Figure 1L) showed only slight improvement. By judging the injury of the gastric mucosa, the study demonstrated that the classified GLPs at doses of high, medium, and low treatment had different degrees of protection on the gastric mucosa of rats. Among them, groups of GLP100-M (Figure 1E), GLP100-H (Figure 1D), and GLP10-L (Figure 1I) have obvious gastroprotective effects. As shown in Appendix A, the model group indicated by an average ulcer index was higher than in other groups. Compared with the model group, the inhibition rate of the ulcer was significantly increased (*p* < 0.05), and all three polysaccharides showed good inhibition efficiency.

### 3.3. H & E Staining of Gastric Mucosa

Figure 2A indicates that the gastric mucosa in a normal group has a complete and clear tissue structure, regular gland arrangements, and no abnormal lesions such as tissue defects and cell infiltration. In Figure 2B, the gastric mucosa of rats showed degeneration and lysis of epithelial cells, lysis and necrosis of mucosal cells, and other gastric injury symptoms in a model group. These symptoms were similar to the study reported by Wang et al. [25], indicating that excessive drinking could directly damage the gastric mucosa. Omeprazole is an effective drug for the treatment of gastric mucosal injury. After omeprazole pretreatment, the gastric mucosa could maintain a complete structure and regular gland arrangements, as shown in Figure 2C, which illustrated that omeprazole has a protective effect on acute alcoholic stomach injury. A comparison of the positive control group, it demonstrated that the classified GLPs had varying degrees of protective effect against alcohol-induced acute gastric injury. After pretreatment with GLP10-M (Figure 2H), GLP100-H (Figure 2J), and GLP100-M (Figure 2K), the gastric mucosa of rats could be better protected without obvious pathological changes. These effects were similar to that of the positive control group. After pretreatment with GLP100-L (Figure 2L), GLP10-L (Figure 2I), GLP1-L (Figure 2F), and GLP1-M (Figure 2E), the symptoms of gastric mucosal injury were slightly relieved, and only epithelial cell degeneration, irregular cell arrangement, and mucosal cytolysis occurred. However, degeneration and lysis of epithelial cells, mucous cell lysis, and submucosal edema still occurred in treatment of GLP1-H and GLP10-H. It can be speculated that the preventive effect of GLPs was achieved by protecting the integrity of mucosal cells and reducing the infiltration of inflammatory cells.

### 3.4. Effects on Serum Antioxidant Indexes of GLPs 

High-concentration alcohol will cause the body to produce a large amount of ROS, which will accelerate the formation and severity of stomach damage [3]. Thus, the ethanol intake will promote oxidative stress in tissues, which will lead to increased lipid peroxidation and decreased antioxidant enzyme activity, thus leading to gastric injury [22]. Antioxidant activity can be used as an indicator to evaluate the resistance and repair ability of the body and gastric tissue to gastric mucosa injury. Previous studies have found that polysaccharides from edible fungi could increase the content of antioxidant enzymes, e.g., SOD, in the body to prevent stomach damage caused by ethanol [22]. As an important free radical scavenging enzyme in the body, SOD can eliminate and neutralize ROS and free radicals, accordingly to protect the gastric mucosa of rats [26]. The protective effects of polysaccharides on gastric mucosal damage could be evaluated based on the content of SOD in rats [27]. In Figure 3A, the content of SOD in each treated group was significantly decreased compared with the blank group (*p* < 0.05). Compared with the model group, the content of SOD was increased in groups of positive drug and classified polysaccharides. It indicated that after being given 75% alcohol gavage, the stomachs were damaged and corresponding oxidative stress reaction occurred. GLPs have protective effects on mucosa in rats, and among them, GLP100-H and GLP100-M had the best protective effects. 

GPX is another important antioxidant enzyme in the body. GPX can prevent red blood cells from being attacked by ROS and thus being hindered from performing their normal function of oxygen transport [28]. Furthermore, we can also judge the degree of protection of the drug against gastric injury by judging the content of GPX [29]. In Figure 3B, the content of GPX was significantly reduced in each treated group (*p* < 0.05). Compared with the model group, groups of omeprazole and classified polysaccharides kept the content of GPX steady. It was demonstrated that GLPs could protect the stomach by maintaining the GPX and reducing the harm of ethanol. Except for the groups of GLP10-M, GLP1-H, and GLP1-L, the effects of other groups on GPX were significantly increased (*p* < 0.05). Among them, GLP100 has the best protective effect on gastric mucosa, and with a dose–response relationship.

### 3.5. Serum Nitric Oxide Content and Epidermal Growth Factor in Rats 

NO is one of the important molecules that lead to pro-inflammatory effects, and many diseases are caused by overproduction of NO [30]. The integrity of the mucus barrier and gastric epithelial cells can be protected by reducing the gastric acid secreted by parietal cells [31]. In physiological conditions, NO acts as an endogenous mediator modulating both the repairing and integrity of the tissues and demonstrates gastroprotective properties against different types of aggressive agents [32]. It has been proved that the protective effect of drugs on alcohol-induced gastric mucosal damage could be evaluated by NO content [25]. In Figure 4A, NO contents in experimental groups were significantly reduced (*p* < 0.05), and the NO contents in the GLP-treated groups were significantly higher than in the model group (*p* < 0.05). The study demonstrated that the content of NO decreased after 75% alcohol gavage, which indicated that alcohol was prone to cause the inflammation. Both the positive drug and classified GLPs could maintain NO content steadily in the groups. Among them, GLP10-M, GLP100-L, and GLP100-M have the best effect.

EGF is secreted in the gastrointestinal tract (not only in epithelial cells, but also in endothelial cells) with the capabilities of reducing secretion of gastric acid, maintaining structural integrity, protecting submucosal blood flow, and speeding up the healing of ulcers [33]. EGF is a very important cytokine produced by the human body, with strong physiological activity, e.g., promoting cell division and keeping the body healthy [25]. A substance that increases the level of EGF in the area of stomach ulcers suggests a strong link in preventing stomach ulcers. Thus, serum EGF content could be used to evaluate the protective effects on the gastric mucosa [34]. In Figure 4B, the EGF content of each treatment group decreased compared with the blank control. Except for the positive drug and GLP100-H, EGF was significantly reduced in all other groups (*p* < 0.05). The EGF content in each treatment group was higher than in the model group. It was indicated that after 75% alcohol gavage, the stomach was damaged and EGF was decreased in serum. GLPs have protective effects on alcoholic stomach injury, in which GLP100-H had the strongest protection ability, while GLP1-H had the weakest ability. Based on the above results, it is speculated that GLPs effectively promoted the secretion of EGF, which is speculated to be the protective mechanism of GLPs on gastric mucosa.

### 3.6. Inflammatory Factors and Interleukin Levels in Rat Serum

MPO is a heme protein contained in neutrophils, which is synthesized in the bone marrow by granulocytes before entering the circulation. External stimuli can cause neutrophils to accumulate and release MPO. Ingestion of high concentrations of alcohol in the stomach will greatly increase the activity of neutrophils, and the number of such cells directly reflects the level of inflammation in the stomach [35]. Ethanol-induced neutrophil infiltration is related to the formation of gastric lesions, which can be determined by MPO activity [5]. MPO can be used to characterize the degree of neutrophil infiltration in damaged tissues. It is used for the quantitative detection of inflammation in experimental gastric injury, colitis, and human gastric ulcer [36]. In Figure 5A, compared with the blank group, MPO increased significantly in all experimental groups except for the GLP100-H-treated group (*p* < 0.05). MPO content in the groups of positive drug and classified polysaccharides decreased compared with the model group. Among them, MPO in groups of GLP100-H and GLP100-M decreased significantly (*p* < 0.05). Accordingly, it was demonstrated that pretreatment with positive drugs and GLPs could inhibit gastric injury.

Ethanol can trigger TLRs, particularly TLR4, which induces the activation of NF-κB, sequentially promoting the production of inflammatory mediators and cytokines, e.g., TNF-α and IL-1β [13]. Once the gastric mucosa is damaged, the inflammatory process is activated to increase inflammatory mediators. Moreover, ethanol can induce cells to overproduce ROS and stimulate macrophages to release inflammatory factors and further activate the NF-κB protein pathway, causing an inflammatory response [1]. Ethanol consumption has been reported to up-regulate pro-inflammatory cytokines, e.g., TNF-α and IL-1β, but down-regulate anti-inflammatory cytokine IL-10 to promote inflammatory response [5]. The gastric damage could be measured by the content of cytokines, such as TNF-α [4]. Once the gastric mucosa is damaged, TNF-α stimulates immune-related cells to produce toxic metabolites. These toxic metabolites can destroy the mucosal self-repair system, thereby delaying mucosal recovery. TNF-α can also aggravate gastric mucosal damage by increasing the release of oxygen free radicals and enhancing the effects of other pro-inflammatory factors. Excessive production of the inflammatory sites of neutrophils trigger oxidative stress and kinase enzymes cause tissue damage in peptic ulcer disease [37,38]. Furthermore, TNF-α, for example, stimulates neutrophil infiltration, IL-1β production, and epithelial cell apoptosis [3]. The pro-inflammatory factor IL-18 and other cytokines jointly induce the TH2-type immune response of the host immune system and worsen the inflammatory injury. In addition, the level of inflammation in patients with gastritis is related to the level of IL-18. IL-18 can stimulate the inflammatory response process through various pathways, thereby enhancing tissue damage. For example, IL-18 can inhibit the maturation of goblet cells and damage the protective mucus layer, thus causing pathological damage to the integrity of the mucosal barrier [39]. Figure 5B-D showed that compared with the blank group, the levels of TNF-α, IL-1β, and IL-18 were significantly increased (*p* < 0.05) in each treatment groups. However, compared with the model group, these levels in the groups of positive drugs and the polysaccharides were significantly reduced (*p* < 0.05). Accordingly, GLPs could significantly inhibit the levels of inflammatory cytokines, in which GLP 100 had the best inhibitory effect. Our results indicated that GLPs could inhibit inflammatory cytokine levels (TNF-α, IL-1β, and IL-18), which implicated the anti-inflammatory effects of GLPs in ethanol-induced gastric injury. This is consistent with a previous study showing that polysaccharides reduce the increase in inflammatory factors induced by ethanol [3,24]. In the future, it is worthwhile to study the effect of polysaccharides on human gastric epithelial cells in vitro to further confirm its effect and mechanism of action.

### 3.7. Histamine Levels in Rat Serum

The gastric mucosa contains a variety of histamines, which are stimulant of gastric acid secretion. Excessive gastric acid secretion would change the permeability of gastric mucosa and accelerate ulcers. Gastric mucosal bleeding and its degree of damage could be judged by identifying the HIS content [40]. In Figure 6, it is indicated that HIS contents in the experimental groups were significantly increased compared with the blank group (*p* < 0.05). HIS contents in the groups of positive drugs and GLPs were significantly lower than that in the model group (*p* < 0.05). Accordingly, omeprazole and GLPs can significantly reduce the HIS content in serum (*p* < 0.05). Among them, the GLP100-H treatment had the most significant effect on reducing the HIS content in serum.

## 4. Conclusions

GLPs were classified by ultrafiltration membranes of 100 kDa, 10 kDa, and 1 kDa with corresponding molecular weights of about 322.0 kDa, 18.8 kDa, and 6.4 kDa, respectively. These classified GLPs had a protective effect on gastric mucosa in rats. In alcohol-induced gastric injury treated by GLPs in rats, antioxidant enzymes, such as SOD, GPX, and MPO, could be regulated, thereby protecting the stomach tissue. GLPs could also maintain the content of signal molecule NO and keep the level of EGF steadily to protect the gastric mucosa from gastric damage. GLPs could inhibit TNF-α, IL-1β, and IL-18 at the level of inflammatory factors, thereby inhibiting the aggravation of gastric injury. All three GLPs can significantly reduce ethanol-induced acute gastric injury in rats and have a certain dose-dependent effect. Moreover, GLP100 has a greater improvement effect on acute gastric injury than GLP10, and GLP10 has a better effect than GLP1. Therefore, the greater the molecular weight of GLPs (6.4~322.0 kDa), the better the effect on acute gastric injury. In addition, GLPs can reduce the serum HIS content in rats with acute gastric injury, which may also contribute to the protective effect of GLPs. Accordingly, different levels and parts of molecular weights of GLPs, especially that above 10 kDa, could reduce alcoholic gastric injury at various degrees. There are still many contents that are worth further study. For instance, further separation and purification of GLPs are needed to determine efficacy. The synergistic effect of the components of each GLP fraction on gastrointestinal protection, and which proportion of GLP components are more effective when mixed in proportion, are worthy of further study, so as to make the research results viable to application.

## Figures and Tables

**Figure 1 nutrients-14-01476-f001:**
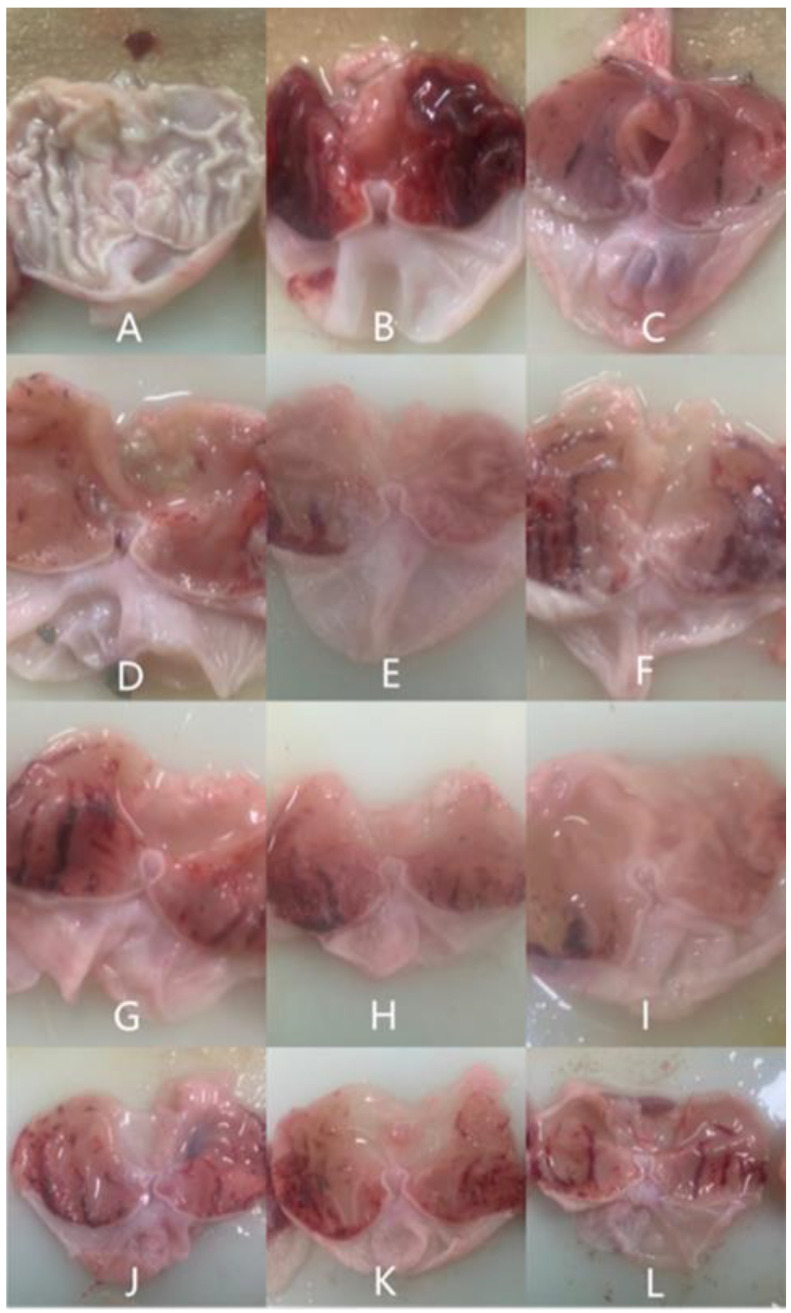
Macroscopic photo images of stomach tissue in rats. Blank group (**A**), Model group (**B**), Positive group (**C**), GLP100-H (**D**), GLP100-M (**E**), GLP100-L (**F**), GLP10-H (**G**), GLP10-M (**H**), GLP10-L (**I**), GLP1-H (**J**), GLP1-M (**K**), GLP1-L (**L**).

**Figure 2 nutrients-14-01476-f002:**
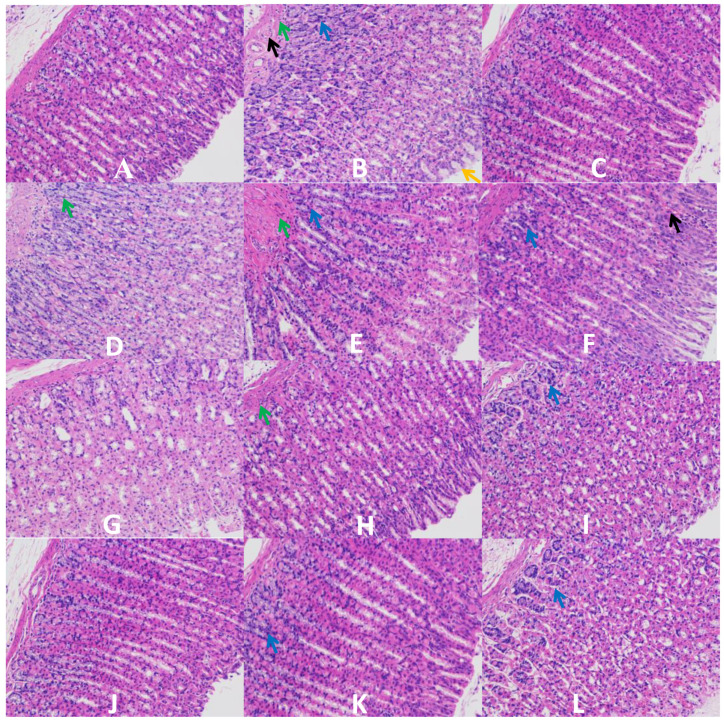
Histological observation of gastric mucosa treated by GLPs. (**A**): blank; (**B**): model; (**C**): control; (**D**): GLP1-H; (**E**): GLP1-M; (**F**): GLP1-L; (**G**): GLP10-H; (**H**): GLP10-M; (**I**): GLP10-L; (**J**): GLP100-H; (**K**): GLP100-M; (**L**): GLP100-L. Black →: vascular congestion; Green →: edema; Blue →: inflammatory cells infiltration; Yellow →: disorganized glandular structure.

**Figure 3 nutrients-14-01476-f003:**
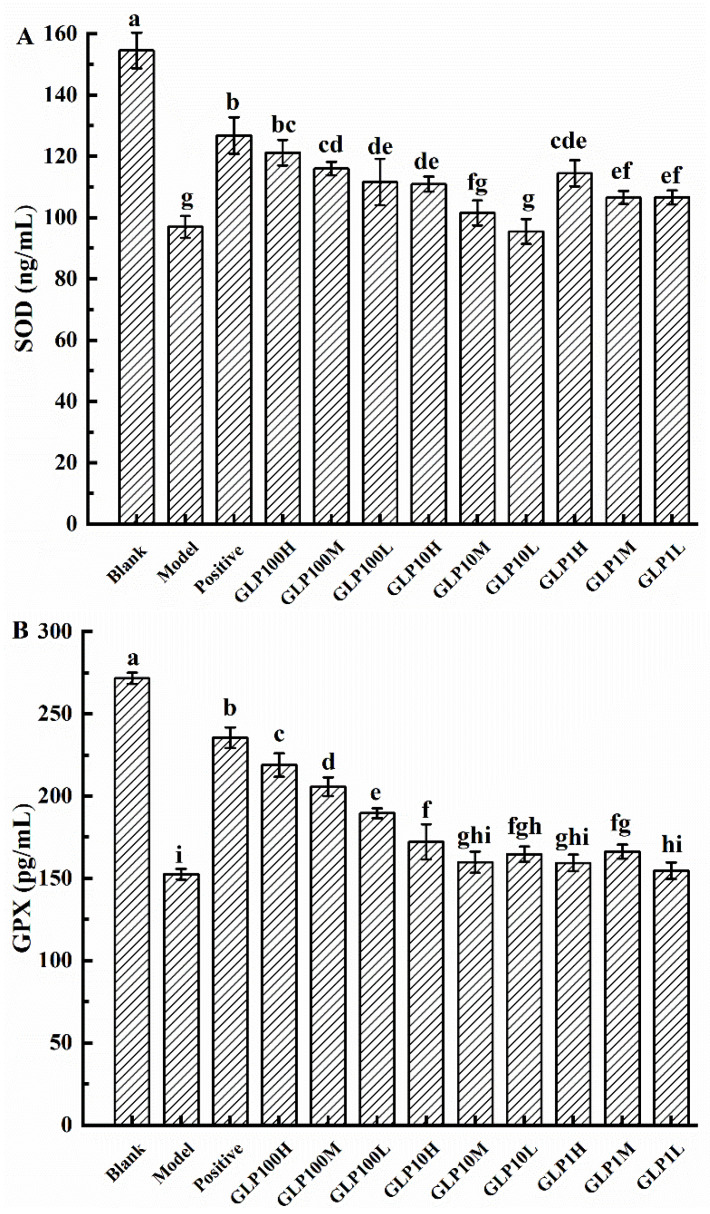
Effects of GLPs on SOD (**A**) and GPX (**B**) activities in rats. Different lowercase letters (a–i) indicate significant differences at *p* < 0.05.

**Figure 4 nutrients-14-01476-f004:**
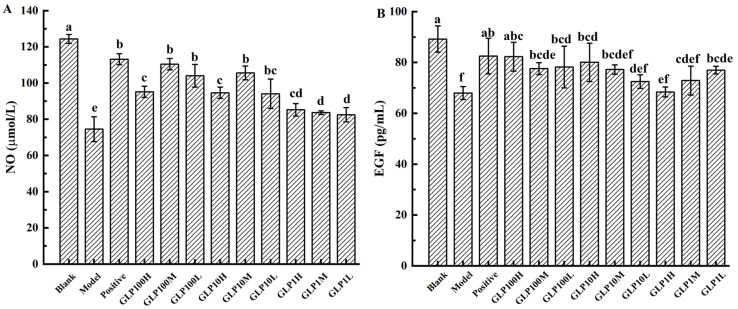
Effects of GLPs on the contents of NO (**A**) and EGF (**B**) in rats. Different lowercase letters (a–e) indicate significant differences at *p* < 0.05.

**Figure 5 nutrients-14-01476-f005:**
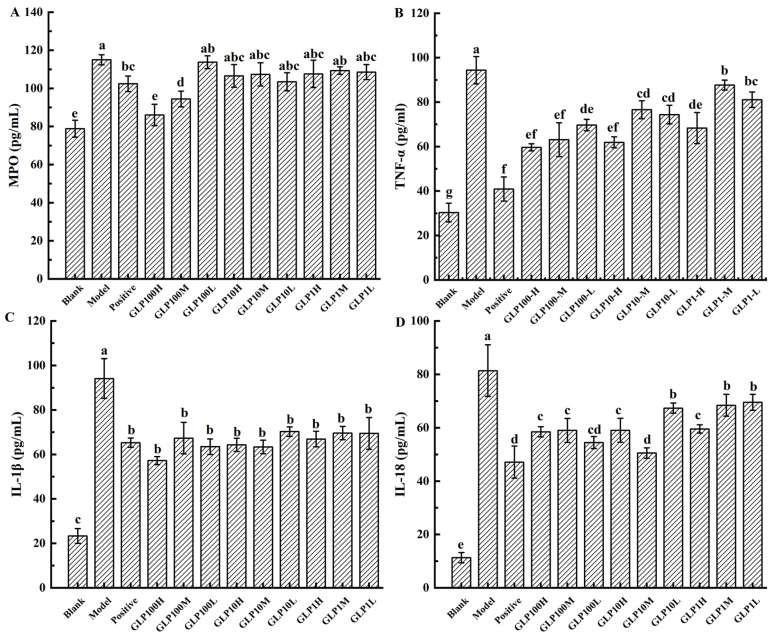
Effect of GLPs on the contents of MPO (**A**), TNF-α (**B**), IL-1β (**C**), and IL-18 (**D**) in rats. Different lowercase letters (a–f) indicate significant differences at *p* < 0.05.

**Figure 6 nutrients-14-01476-f006:**
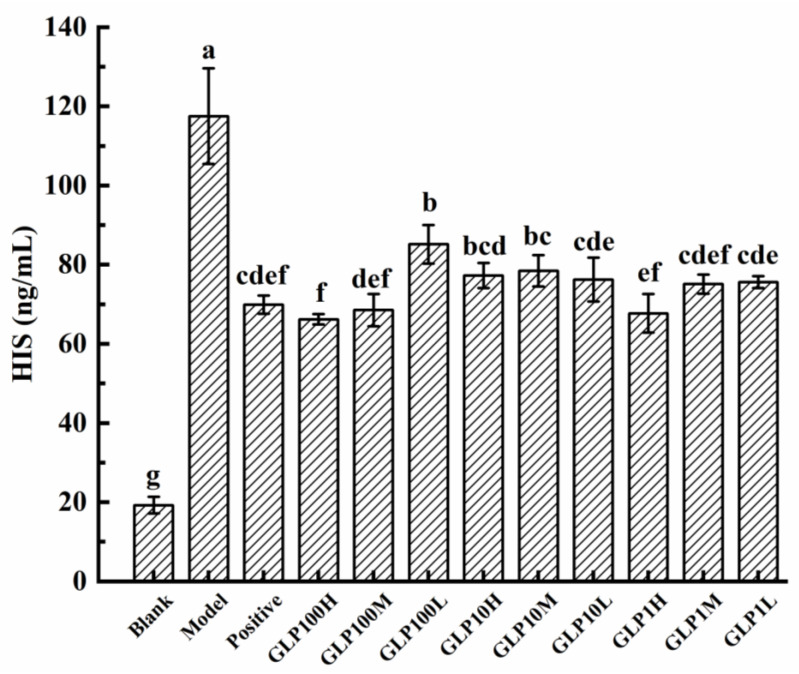
Effects of GLPs on HIS content in rats. Different lowercase letters (a–g) indicate significant differences at *p* < 0.05.

## Data Availability

Further data in this study are available on request from the author.

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
