# Peer review of "Gastroprotective Effects of Ganoderma lucidum Polysaccharides with Different Molecular Weights on Ethanol-Induced Acute Gastric Injury in Rats"

_nutrients, 2022, doi:10.3390/nu14071476_

Round 1

Reviewer 1 Report

The present manuscript may be recommended for publication after some revisions. All the authors should recheck all the manuscript and revise all the small errors and grammatical mistakes. For example, the following points should be reconsider:

1. L22-25: Results showed that GLP with different molecular weights had a dose-dependent effect in alleviating alcohol-induced gastric injury. The underlying mechanism might be related to regulating anti-oxidation, promoting the release of related defense factors, reducing inflammatory factors, and inhibiting histamine release. 

2. L199: This section may be divided into subheadings.   

3.  L217-218Gastric mucosal injury is characterized by gastric mucosal erosion, extending to the muscle layer to puncture the gastric wall [4].   

4. L304-307: It demonstrated that GLPs could protect the stomach by maintaining the 304 GPX, reducing the harm of ethanol. Except for the groups (of) GLP10-M, GLP1-H, and GLP1-L, the effects of other groups on GPX were significant (P<0.05).  

 5. L319-321: In Fig. 4B, the EGF content of each treatment group decreased compared with the blank control. Except for the (groups of) positive drug and GLP100-H, EGF was significantly decreased (reduced) (P<0.05) in all other groups. EGF  content in each drug group was higher than (that) in the model group.

6.  In the discussion section, it would be more informative to include a clear explanation on the synergetic effects of the components of each GLPs fractions. If the authors can define in  which ratio of the constituent may be more effective. The observed protective results are more sensitive on which constituents or combinations. 

Author Response

Response to Reviewer 1 Comments

The present manuscript may be recommended for publication after some revisions. All the authors should recheck all the manuscript and revise all the small errors and grammatical mistakes. For example, the following points should be reconsider:

Response: Thank you very much for your helpful comments and giving us the opportunity for further revisions. It is very gratified that the reviewer had a positive attitude towards our research. We have tried our best to improve the manuscript and made some changes in the manuscripts based on your comments.

Point 1: 1. L22-25: Results showed that GLP with different molecular weights had a dose-dependent effect in alleviating alcohol-induced gastric injury. The underlying mechanism might be related to regulating anti-oxidation, promoting the release of related defense factors, reducing inflammatory factors, and inhibiting histamine release. 

Response 1: Thank you very much for your comments. We revised and improved the abstract according to the comments of the reviewers. Please refer to the tracking record for the modified part.

Point 2: 2. L199: This section may be divided into subheadings.  

Response 2: Thank you very much for your insightful comments. This section has been divided into three subheadings.

Point 3: 3.  L217-218: Gastric mucosal injury is characterized by gastric mucosal erosion, extending to the muscle layer to puncture the gastric wall [4].   

Response 3: Thanks for reviewer’s valuable comments. It has been modified according to the guidance of the author.

Point 4: 4. L304-307: It demonstrated that GLPs could protect the stomach by maintaining the 304 GPX, reducing the harm of ethanol. Except for the groups (of) GLP10-M, GLP1-H, and GLP1-L, the effects of other groups on GPX were significant (P<0.05).  

Response 4: Thanks for reviewer’s valuable comments. We have revised the manuscript according to the reviewer’s comments.

Point 5:  5. L319-321: In Fig. 4B, the EGF content of each treatment group decreased compared with the blank control. Except for the (groups of) positive drug and GLP100-H, EGF was significantly decreased (reduced) (P<0.05) in all other groups. EGF  content in each drug group was higher than (that) in the model group.

Response 5: Thanks to reviewers for their careful and thoughtful comments. We have revised the manuscript according to the reviewer’s comments.

Point 6: 6.  In the discussion section, it would be more informative to include a clear explanation on the synergetic effects of the components of each GLPs fractions. If the authors can define in which ratio of the constituent may be more effective. The observed protective results are more sensitive on which constituents or combinations. 

Response 6: Thanks for reviewer’s valuable comments. We have revised the manuscript according to the reviewer's comments. Accumulating studies confirmed the protective effects of natural polysaccharides on gastric ulcers and gastritis. Polysaccharides of Dendrobium officinale Kimura & Migo could protect the gastric mucosa by inhibiting cell apoptosis induced by oxidative stress [1]. Polysaccharides of Momordica charantia could reduce the ethanol induced gastric injury in rats by inhibiting gastritis and oxidative stress [2]. Accordingly, polysaccharides could be as an effective treatment or health supplement for gastric mucosal diseases. GLPs protected against ethanol-induced acute liver injury in vivo and in vitro via the TLR4/NF-κB signaling pathway, obviously reduced the secretion of inflammatory cytokines (e.g., TNF-α and IL-1β) [3]. Polysaccharides from Dendrobium nobile could alleviate ethanol-induced histological damage, antioxidant activities, the level of epidermal growth factor (EGF) [4]. Our preliminary experiments also found that GLPs can improve the injury of acute gastroenteritis, but we do not know which components play a role, whether it is related to molecular weight, so we conducted the study in this paper. Membranes can be used to classify polysaccharides according to their molecular weights. Cascade membrane technology with ultrafiltration membranes of 100 kDa, 10 kDa and 1 kDa has been used to classify the polysaccharides of Ganoderma lucidum at different molecular weights. Based on our collected data, we report for the first time that GLPs ameliorates alcohol-induced acute gastric injury by molecular weight. All three GLPs can significantly reduce the acute gastric injury induced by ethanol in rats and have a certain dose-dependent effect. Moreover, GLP100 has a greater improvement effect on acute gastric injury than GLP10, and GLP10 has a better effect than GLP1. Therefore, the greater the molecular weight of GLPs (6.4 ~ 322.0 kDa), the better the effect on acute gastric injury.

Thanks to the reviewers for guiding our future research. There are still a lot of contents worth further study. For instance, further separation and purification of GLPs is needed to determine its efficacy. The synergistic effect of the components of each GLP fractions on gastrointestinal protection and which proportion of GLP components are more effective when mixed in proportion, are worthy of further study, so as to make the research results towards application. These are the kinds of things we plan to do. We have written all this down in the manuscript.

Our current research is only at the preliminary research stage and is limited to the separation of GLPs by molecular weight. Each component in GLPs has been isolated and purified, and the study of the function and mechanism of each component is our upcoming work. Next, we will investigate the mechanism of action of GLPs in cellular level, and use a randomized controlled trial to study the effect of ameliorating acute gastric injur of GLPs in population-based studies. More research is needed to develop specific dietary recommendations, and this article is just the beginning.

References

  1. Zeng, Q.; Ko, C.H.; Siu, W.S.; Li, L.F.; Han, X.Q.; Yang, L.; Bik-San Lau, C.; Hu, J.M.; Leung, P.C. Polysaccharides of Dendrobium officinale Kimura & Migo protect gastric mucosal cell against oxidative damage-induced apoptosis in vitro and in vivo. J Ethnopharmacol 2017, 208, 214-224, doi:10.1016/j.jep.2017.07.006.
  2. Raish, M.; Ahmad, A.; Ansari, M.A.; Alkharfy, K.M.; Aljenoobi, F.I.; Jan, B.L.; Al-Mohizea, A.M.; Khan, A.; Ali, N. Momordica charantia polysaccharides ameliorate oxidative stress, inflammation, and apoptosis in ethanol-induced gastritis in mucosa through NF-kB signaling pathway inhibition. Int J Biol Macromol 2018, 111, 193-199, doi:10.1016/j.ijbiomac.2018.01.008.
  3. Yang, K.; Zhan, L.; Lu, T.; Zhou, C.; Chen, X.; Dong, Y.; Lv, G.; Chen, S. Dendrobium officinale polysaccharides protected against ethanol-induced acute liver injury in vivo and in vitro via the TLR4/NF-κB signaling pathway. Cytokine 2020, 130, 155058, doi:https://doi.org/10.1016/j.cyto.2020.155058.
  4. Zhang, Y.; Wang, H.; Mei, N.; Ma, C.; Lou, Z.; Lv, W.; He, G. Protective effects of polysaccharide from Dendrobium nobile against ethanol-induced gastric damage in rats. International Journal of Biological Macromolecules 2018, 107, 230-235, doi:https://doi.org/10.1016/j.ijbiomac.2017.08.175.

Reviewer 2 Report

The manuscript is interesting and well organized.

Below I present some minor remarks:

L 86 lentinus - correct to Lentinus

L 150 Athanol - correct to Ethanol

L 199-201 This part should be removed.

Suppemenatry: Figure 1 - correct to Figure S1

power of Gludium ?? - it should be powder of G. lucidum?

Author Response

Response to Reviewer 2 Comments

The manuscript is interesting and well organized.

Response: Thank you very much for your helpful comments and giving us the opportunity for revisions. We have tried our best to improve the manuscript according to your guidance. My co-authors and I sincerely hope that the revision can be satisfied with you.

Point 1:

Below I present some minor remarks:

L 86 lentinus - correct to Lentinus

L 150 Athanol - correct to Ethanol

Response 1: Thanks for reviewer’s valuable comments. We have revised some mistakes raised by reviewers and examined in detail other parts in the manuscript.

Point 2: L 199-201 This part should be removed.

Response 2: Thank you very much for your insightful comments. We have removed the extra parts raised by reviewers and examined in detail other parts of the paper.

Point 3:

Suppemenatry: Figure 1 - correct to Figure S1

power of Gludium ?? - it should be powder of G. lucidum?

Response 3: We are very sorry for the incorrect spelling. We have revised the mistakes raised by reviewers and examined in detail other parts of the paper

Reviewer 3 Report

This article reported the gastroprotective property of GLPs against ethanol-induced acute gastric injury and the underlying mechanism for it.

However, reviewer thinks that “method” or “discussion” seems to contain some kind of mistake. 

Therefore, the following point would be considered to improve this article.

Major point

1) Authors described that omeprazole and GLPs have significantly inhibitory effects on histamine release in line 404-405.

However, authors measured histamine contents in serum.

Which tissue or organ or cells dose histamine release from ?

Dose the histamine contents in serum change by the inhibiting histamine release in the stomach ?

2) In addition, authors described GLPs also can inhibit the release of gastric acid by inhibiting the release of histamine in conclusion (line418) and in abstract (line25).

Why can authors estimate the property of GLPs as inhibiting histamine release only by serum contents of histamine ?

Minor point

1) The description is not correct.

In “Materials and Methods” section

between line174 and line175

      Ulcer inhibitory rate %  =  (1- *********) x100

 If authors want to use percentage notation, it is needed to multiply by 100.

2) The expression style should be unified.

Ex)

In “Materials and Methods” section

Line123 : 700 r/min (?) or   Line183 : 6000 rpm (?)

In “References” section

Ref No.1 : International Journal of Biological Macromolecules  → Int J Biol Macromol

No.5 : International Journal of Biological Macromolecules  → Int J Biol Macromol

No.14 : Journal of the Taiwan Institute of Chemical Engineers  → J Taiwan Inst Chem Eng

No.15 : Trends in Food Science & Technology  →   Trends Food Sci Technol

No.22 : Food Chemistry  →   Food Chem

No.32 : International Journal of Pharmacology  →   Int J Pharmacol

No.33 : Biomedicine & Pharmacotherapy  →   Biomed Pharmacother

No.39 : Food and Chemical Toxicology  →   Food Chem Toxicol

Author Response

Response to Reviewer 3 Comments

Point 1:

This article reported the gastroprotective property of GLPs against ethanol-induced acute gastric injury and the underlying mechanism for it.

However, reviewer thinks that “method” or “discussion” seems to contain some kind of mistake. 

Therefore, the following point would be considered to improve this article.

Response 1: Thanks for reviewer’s valuable comments. We have tried our best to improve the manuscript according to your guidance. My co-authors and I sincerely hope that the revision can be satisfied with you.

Point 2: Major point

1) Authors described that omeprazole and GLPs have significantly inhibitory effects on histamine release in line 404-405.

However, authors measured histamine contents in serum.

Which tissue or organ or cells dose histamine release from ?

Dose the histamine contents in serum change by the inhibiting histamine release in the stomach?

Response 2: Thank you very much for your insightful comments. Based on the advice of the reviewers, we have changed this part of the content.

Histamine, also known as 2-imidazol-4-ylethylamine for C5H9N3, is stimulated by the mast cells and basophils degranulation released after histidine, treats with decarboxylase, widely exists in the human body each organization mucosa. In the resting state, histamine is stored in the particles of mast cells as shown by protein complexes. When the body is injured, histamine can be activated and released in a free state, targeting specific histamine receptors to play a physiological role [1,2].  Histamine can also be produced by basophils and other immune cells [3] but much higher concentrations of histamine may be found in intestinal mucosa, skin, and bronchial tissues. Histamine regulates a plethora of pathophysiological and physiological processes, such as secretion of gastric acid, inflammation, and the regulation of vasodilatation and bronchoconstriction [4]. In addition, it can also serve as a neurotransmitter [4]. Histamine is involved in a wide variety of pathological and physiological processes including allergic reactions, inflammation, immune responses, gastric secretion, and neurotransmission. Intracellularly synthetized histamine is stored in cytoplasmic granules of mast cells and basophils, two of the major producers of histamine [2].The change of serum histamine content can reflect the release of histamine.

Histamine, a major mediator released by mast cells, contributes to increasing vascular permeability of the gastrointestinal mucosa [5]. Increased production of histamine induces increases in gastric blood flow, secretion of gastric acid and gastric mucus production with promoting gastroprotection and gastric ulcer healing [6]. All these studies indicate that increased formation of histamine or exogenous histamine or its anologs have been accompanied with gastroprotection and acceleration of gastric ulcer healing [6]. A previous experimental study has shown that treatment with histamine protects the gastric mucosa and enhances the gastroprotective effect of ranitidine and omeprazole against stress-induced gastric lesions [7]. Histamine alone reduced ulcer area evoked by stress and this effect was accompanied by an increase in gastric mucosal blood flow and mucosal DNA synthesis, as well as a decrease in serum pro-inflammatory interleukin-1β concentration [8]. Histamine exhibits protective effect against stress-induced gastric ulcer and that this gastroprotection is related to stimulation of histamine H1 and H3 receptors [8]. Previous similar studies measured changes in serum and stomach histamine levels, which tended to increase or decrease in the same way [6]. In summary, we believe that the improvement effect of GLPs on acute gastroenteritis may be related to the release of histamine.

Of note, the relationship between the change of serum histamine content and the release of histamine in stomach deserves further study. According to the comments of the experts, we also understand that the histamine level in serum is increased, but we can't speculate that the histamine level in stomach tissue must be increased. We have made corresponding modifications to the expression of the manuscript content. Based on the existing literature reports, we made corresponding reasonable speculations. 

Point 3:

2) In addition, authors described GLPs also can inhibit the release of gastric acid by inhibiting the release of histamine in conclusion (line 418) and in abstract (line 25).

Why can authors estimate the property of GLPs as inhibiting histamine release only by serum contents of histamine ?

Response 3: Thank you very much for your helpful comments. After reading and understanding this manuscript again, there is no strong evidence that GLP can inhibit histamine release. Serum histamine content to estimate the properties of GLPs for inhibiting histamine release is inadequate, and we add the possibility of expression. The reasons are as follows.

Histamine, a major mediator released by mast cells, contributes to increasing vascular permeability of the gastrointestinal mucosa [5]. Increased production of histamine induces increases in gastric blood flow, secretion of gastric acid and gastric mucus production with promoting gastroprotection and gastric ulcer healing [6]. All these studies indicate that increased formation of histamine or exogenous histamine or its anologs have been accompanied with gastroprotection and acceleration of gastric ulcer healing [6]. A previous experimental study has shown that treatment with histamine protects the gastric mucosa and enhances the gastroprotective effect of ranitidine and omeprazole against stress-induced gastric lesions[7]. Histamine alone reduced ulcer area evoked by stress and this effect was accompanied by an increase in gastric mucosal blood flow and mucosal DNA synthesis, as well as a decrease in serum pro-inflammatory interleukin-1β concentration [8]. Histamine exhibits protective effect against stress-induced gastric ulcer and that this gastroprotection is related to stimulation of histamine H1 and H3 receptors [8]. Previous similar studies measured changes in serum and stomach histamine levels, which tended to increase or decrease in the same way [6]. In summary, we believe that the improvement effect of GLPs on acute gastroenteritis may be related to the release of histamine.

Of note, the relationship between the change of serum histamine content and the release of histamine in stomach deserves further study. According to the comments of the experts, we also understand that the histamine level in serum is increased, but we can't speculate that the histamine level in stomach tissue must be increased. We have made corresponding modifications to the expression of the manuscript content. Based on the existing literature reports, we made corresponding reasonable speculations. 

Point 4:

Minor point

1) The description is not correct.

In “Materials and Methods” section

between line 174 and line 175

      Ulcer inhibitory rate %  =  (1- *********) x100

 If authors want to use percentage notation, it is needed to multiply by 100.

Response 4:  Thanks for reviewer’s valuable comments. We have revised the mistake raised by reviewers and examined in detail other parts of the paper.

Point 5: 2) The expression style should be unified.

Ex) In “Materials and Methods” section

Line 123 : 700 r/min (?) or   Line183 : 6000 rpm (?)

Response 5: Thank you for this insightful comment. We have unified the units in the text and examined in detail other parts of the paper.

Point 6: In “References” section

Ref No.1 : International Journal of Biological Macromolecules  → Int J Biol Macromol

No.5 : International Journal of Biological Macromolecules  → Int J Biol Macromol

No.14 : Journal of the Taiwan Institute of Chemical Engineers  → J Taiwan Inst Chem Eng

No.15 : Trends in Food Science & Technology  →   Trends Food Sci Technol

No.22 : Food Chemistry  →   Food Chem

Response 6: Thank the reviewers for their constructive guidance in our study. We have revised the references in the manuscript. The modified words have been marked by using tracking text in the manuscript.

References

  1. Li, P.; Zhou, B.; Ge, M.; Jing, X.; Yang, L. Metal coordination induced SERS nanoprobe for sensitive and selective detection of histamine in serum. Talanta 2022, 237, 122913, doi:https://doi.org/10.1016/j.talanta.2021.122913.
  2. Slamet Soetanto, T.; Liu, S.; Sahid, M.N.A.; Toyama, K.; Maeyama, K.; Mogi, M. Histamine uptake mediated by plasma membrane monoamine transporter and organic cation transporters in rat mast cell lines. European Journal of Pharmacology 2019, 849, 75-83, doi:https://doi.org/10.1016/j.ejphar.2019.01.050.
  3. Saluja, R.; Ketelaar, M.E.; Hawro, T.; Church, M.K.; Maurer, M.; Nawijn, M.C. The role of the IL-33/IL-1RL1 axis in mast cell and basophil activation in allergic disorders. Molecular Immunology 2015, 63, 80-85, doi:https://doi.org/10.1016/j.molimm.2014.06.018.
  4. Thangam, E.B.; Jemima, E.A.; Singh, H.; Baig, M.S.; Khan, M.; Mathias, C.B.; Church, M.K.; Saluja, R. The Role of Histamine and Histamine Receptors in Mast Cell-Mediated Allergy and Inflammation: The Hunt for New Therapeutic Targets. Frontiers in Immunology 2018, 9, doi:10.3389/fimmu.2018.01873.
  5. Sun, P.; Li, D.; Li, Z.; Dong, B.; Wang, F. Effects of glycinin on IgE-mediated increase of mast cell numbers and histamine release in the small intestine. The Journal of Nutritional Biochemistry 2008, 19, 627-633, doi:https://doi.org/10.1016/j.jnutbio.2007.08.007.
  6. Erkasap, N.; Uzuner, K.; Serteser, M.; Köken, T.; Aydın, Y. Gastroprotective effect of leptin on gastric mucosal injury induced by ischemia–reperfusion is related to gastric histamine content in rats. Peptides 2003, 24, 1181-1187, doi:https://doi.org/10.1016/j.peptides.2003.06.007.
  7. Warzecha, Z.; Dembiński, A.; Brzozowski, T.; Ceranowicz, P.; Dembiński, M.; Stachura, J.; Konturek, S.J. Histamine in stress ulcer prophylaxis in rats. Journal of Physiology and Pharmacology 2001, 52, 407-421.
  8. Dembiński, A.; Warzecha, Z.; Ceranowicz, P.; Brzozowski, T.; Dembiński, M.; Konturek, S.J.; Pawlik, W.W. Role of capsaicin-sensitive nerves and histamine H1, H2, and H3 receptors in the gastroprotective effect of histamine against stress ulcers in rats. European Journal of Pharmacology 2005, 508, 211-221, doi:https://doi.org/10.1016/j.ejphar.2004.11.059.

Round 2

Reviewer 1 Report

The present manuscript may be recommended for publication.

Author Response

Thanks for the reviewer's recognition of our manuscript publication, thanks again for the reviewer's contribution to the revision!

Reviewer 3 Report

Reviewer confirmed that the all things I pointed out have been corrected appropriately, except for one point described in the lower row. In the article, reference No41, Piqueras et. al. administrated histamine by intravenous injection. Reviewer can understand that the administrated histamine could act at the stomach, although histamine is one of an autacoid. However, authors judged the effects of GLPs as “histamine release” in serum in Figure 6. Where is the origin of the histamine in serum ? Is the origin mast cells in stomach or basophils in blood ? Reviewer thinks that authors assume the gastric mucosa as a target of GLPs. In that case, dose the histamine contents in serum increase if the contents of histamine increases in stomach according to increasing the histamine release from mast cells ? Reviewer thinks it can’t be evaluated the histamine release by the histamine contents in serum. The expression should be changed.

Author Response

Response to Reviewer 3 Comments

Point: Reviewer confirmed that the all things I pointed out have been corrected appropriately, except for one point described in the lower row. In the article, reference No41, Piqueras et. al. administrated histamine by intravenous injection. Reviewer can understand that the administrated histamine could act at the stomach, although histamine is one of an autacoid. However, authors judged the effects of GLPs as “histamine release” in serum in Figure 6. Where is the origin of the histamine in serum? Is the origin mast cells in stomach or basophils in blood? Reviewer thinks that authors assume the gastric mucosa as a target of GLPs. In that case, dose the histamine contents in serum increase if the contents of histamine increases in stomach according to increasing the histamine release from mast cells ? Reviewer thinks it can’t be evaluated the histamine release by the histamine contents in serum. The expression should be changed.

Response: Thanks for reviewer’s valuable comments. We have tried our best to improve the expression in the manuscript according to your guidance. The modified words have been marked by using tracked text in the manuscript.

The changes are as follows in the manuscript:

  1. The underlying mechanism might be related to regulating anti-oxidation, promoting the release of related defense factors, reducing inflammatory factors, and reducing the level of histamine in serum.
  2. Accordingly, omeprazole and GLPs can significant reduce the histamine content in serum (P < 0.05). Among them, the GLP100-H treatment had the most significant effect on reducing the histamine content in serum.
  3. In addition, GLPs can reduce the serum HIS content in rats with acute gastric injury, which may also contribute to the protective effect of GLPs.
